# Influence of Child’s Temperament on Behaviour Management Problems in the Dental Office: A Literature Review

**DOI:** 10.3390/children10010090

**Published:** 2023-01-02

**Authors:** Nhat Minh Do, François Clauss, Margot Schmitt, Marie-Cécile Manière

**Affiliations:** Department of Paediatric Dentistry, Faculty of Dental Surgery of Strasbourg, University of Strasbourg, 67000 Strasbourg, France

**Keywords:** child, temperament, dental behaviour management problems (DBMP), dental anxiety

## Abstract

Background: A child’s temperament could have an influence on his/her behaviour in the dental environment. This review aims to present the main temperament surveys and their clinical use and to discuss the relationship between certain temperament dimensions and Dental Behaviour Management Problems (DBMP). Methods: A literature search was conducted in Medline/PubMed, ScienceDirect, Wiley Online Library and Cochrane library electronic databases for publications, up to June 2022, investigating the link between child’s temperament and DBMP. Results: From 733 potentially eligible studies, 12 were included in qualitative synthesis. Conclusion: According to studies using the Child Behaviour Questionnaire (CBQ) scale, the most impactful dimensions are activity, extraversion and surgency, high-intensity pleasure and attention control. For those using the Emotionality–Activity–Sociability (EAS) scale, emotionality and shyness have a statistically significant positive linear correlation with dental anxiety and DBMP. It has yet to be determined whether the use and interpretation of these questionnaires can be carried out in a daily clinical situation as an aid to sharpen the indications for the several levels of sedation.

## 1. Introduction

Paediatric dentists are considered experts in the care of children with disruptive behaviours in the dental office. This phenomenon, defined as Dental Behaviour Management Problems (DBMP), combines the restlessness and lack of cooperation of the child, leading to the interruption of an intervention and deferred care. According to research, this issue affects between 10% to 20% of children, creating a real public health problem [1,2].

Cognitive therapies and behavioural approaches are often insufficient to calm these children. Consequently, there are different levels of sedation that dentists can use. The selection and the indication for a pharmacological sedation technique are often empirical and depend on both a subjective evaluation and the experience of the dentist, rather than on a reproducible method.

The American Academy of Paediatric Dentistry (AAPD) reports a failure rate of 27% for moderate conscious sedation. The main cause of failure is the child’s behaviour during a session, which prevents the continuation of the treatment [3]. Therefore, it may be helpful to identify children with DBMP and to predict their behaviour before the intervention in order to present them with a suitable sedation method.

The search for factors known to increase a child’s risk for maladjustment to stress and his/her resilience can be focused on the child’s attributes. One lead could be the analysis of the temperament and character of the child, which are inherent to his/her personality.

Although the notion of temperament is not commonly agreed upon, it is generally referred to as a “set of stable, heritable and relatively independent traits having neurophysiological and neurobiological correlations, of which combination makes it possible to characterise individuals” [4]. In other words, temperaments are individual differences in emotional responsiveness and behaviour in response to a particular situation. The child’s personality would result, in part, from the combination of temperament and environmental determinants such as learning.

Indeed, the hereditary characteristics of an individual’s nature have been found to be a useful and notable predictor of how a child would react to a variety of potentially stressful situations. The child’s temperament has been shown to impact his/her anxiety levels and coping mechanism to stress [5,6,7]. Smith [5] and Prior [5] have observed that the teachers’ ratings of positive temperament, such as low emotional reactivity and high social engagement, best discriminated children showing resilience to stress [5]. For this study, several questionnaires were used to assess the children’s characteristics, including the Emotionality, Activity, Sociability (EAS) temperament scale created by Buss and Plomin [8] (1984).

The objective of this article was to evaluate the usage of temperamental surveys and to discuss the relationship between certain temperament dimensions and Dental Behaviour Management Problems (DBMP). Their interpretation in the context of paediatric dentistry could be useful in order to identify children with more risks of DBMP.

## 2. Materials and Methods

The review was conducted in accordance with the Preferred Reporting Items for Systematic Reviews and Meta-Analyses (PRISMA) guidelines [9].

### 2.1. Search Strategy

A literature search in the electronic databases Medline/PubMed and Cochrane was performed up to June 2022 with the following keywords: temperament, child behaviour and dental environment. In addition, the bibliographies of included articles were scanned manually to identify any additional relevant articles. The search strategy is shown in Table 1.

The search was updated and extended to several other databases—ScienceDirect and Wiley Online Library—in November 2022 to identify new studies published since the original search was conducted.

### 2.2. Eligibility Criteria

To be included in the literature review, the predetermined eligibility criteria were as follows: (a) observational cross-sectional and cohort studies; (b) published in English or French in a peer-reviewed journal; (c) dealing with the predictive nature of temperament in children; (d) conducted in children up to 16 years old; (e) should address the behaviour during dental interventions.

There was no time restriction.

Two researchers (NMD and MS) independently screened all titles and abstracts. Discrepancies were resolved by collegiate decision with the other authors. Following selection, a full-text analysis was performed independently by two researchers (NMD AND MS).

### 2.3. Data Collected

Data extraction was independently conducted by two researchers (NMD AND MS). A third researcher (FC) confirmed the accuracy of the collected data.

The following data were collected from the included case reports and case series: time of publication, number of patients, age of the patients, temperament scale used, type of procedure and sedation, correlation of temperament and finally the behavioural rating used and the temperament, which are statistically associated with DBMP.

## 3. Results

### 3.1. Literature Search

The performed literature search strategy initially retrieved a total of 914 articles, of which 733 remained after removal of duplicates.

Among the 733 articles selected, only 12 matched the inclusion criteria.

The results of the screening and search process are presented in Figure 1.

### 3.2. Descriptive Analysis

A summary of included articles is reported in Table 2.

Several temperament models exist, and the main models used in the included articles are the BSQ and the EAS scales.

#### 3.2.1. The Behaviour Style Questionnaire (BSQ) Scale

The BSQ was first presented by Thomas and Chess [22], who isolated nine dimensions of temperament:**Activity**: the level of motor activity and the amount of time the child is active.**Rhythmicity/regularity**: patterns of eating, sleeping and other bodily functions.**Approach and the withdrawal**: ease of approaching people and situations.**Adaptability**: the way the child responds to changes in his/her environment.**Response**: the energy levels and intensity of the child’s response.**Disposition**: predominant quality of mood.**Sensitivity**: the threshold for stimuli.**Distractibility**: how easily the child can be distracted from what he is doing.**Attention span/persistence**: the span of time for which the child will pay attention to one thing when left to his/her own devices and their persistence with an activity. [22]

From their observations, these two authors distinguished several types of children: **easy children, difficult children and those with slow adaptation**.

This model made it possible to construct one of the most widely used temperament scales: the “Behaviour Style Questionnaire”, also known as the “Toddler Behaviour Assessment Questionnaire”, for children aged between 15 and 36 months and the “Child Behaviour Check List” from 4 to 18 years old, whereas the French study was conducted with boys aged between 6 and 11 [23].

Other, similar scales exist, such as the “Early Childhood Behaviour Questionnaire” for ages 18 to 36 months, as well as the “Childhood Behaviour Questionnaire” for ages 3 to 7. These questionnaires can be time consuming, since they contain more than 100 items addressing the nine dimensions of temperament.

Of the 12 articles included in this review, 6 used the BSQ or a similar scale (TTS, CBQ).

#### 3.2.2. The Emotionality–Activity–Sociability (EAS) Scale

Models that are more recent have criticised the lack of validity of these nine dimensions, which are not truly independent of each other. Among these recent models, Buss and Plomin [8] presented four dimensions:**Emotionality, distress**: in the face of a threatening event, variations on this dimension can range from a stoic lack of reaction to distress beyond the child’s emotional control.**Activity**: the pace and energy of the child.**Sociability**: the desire to interact with the social environment.Impulsiveness is replaced by Shyness due to its lower-than-expected level of heritability. **Shyness** refers to inhibiting behaviours in the presence of strangers and a tendency to shy away from social interactions.

Of the 12 articles included in this review, 4 used the EAS scale.

### 3.3. Child Behaviour in a Dental Environment

In the dental environment, we found heterogeneous studies both in terms of the child’s inclusion age, and in terms of the protocol applied, which complicates the interpretation.

#### 3.3.1. During the First Session with the Dentist

Some authors evaluated the predictive power of the child’s temperament on his/her behaviour during the first session with the dental surgeon (without sedation, and with non-invasive [11,17] and invasive [15,17] procedures). It can be seen that neither the temperament nor the behavioural scales are the same in these three articles, which highlights the need for standardisation of the scales. Despite the disparity of these scales and the included age groups, which prevented a reliable comparison of results between studies, several statistical trends can be observed. In fact, shyness (*p* = 0.002 for Radis [11], *p* < 0.003 for Tsoi [19] and Jain [20]), associability (*p* = 0.00001 for Pai [17]) and emotionality (*p* < 0.001 for Tsoi [19]) seem to be linked with children who are difficult to approach during their first visits. The hyperactivity seems to provide more significant results in the second step of regression, as suggested by the study of Radis [11] and Tsoi [19] (*p* < 0.03).

#### 3.3.2. During Invasive Procedures

Aminabadi [15] (2011) reported a link between temperament traits: anger, irritability, fear, reactivity, reaction and shyness with extremely negative behaviours (*p* < 0.05) when performing restoration under local anaesthesia. Likewise, during an invasive dental procedure, Janeshin [21] (2021) found a correlation between anger and perceptual sensitivity which was higher in patients with extremely negative behaviour (*p* = 0.004 and *p* = 0.001, respectively) than other temperament traits.

#### 3.3.3. Patients Referred for Sedations

The same study pattern can be found in children referred for anxiety or DBMP for management under various pharmacological methods of sedation. We noted failure rates (between 10% and 20%) equivalent to or lower than those reported by the AAPD.

The same temperament traits significantly associated with failed DBMP sedation are hyperactivity (*p* = 0.004 for Nelson [18], negative emotionality (in the form of mood swings) (*p* = 0.01) for Jensen [13], frustration, anger, and the child’s ability to be comforted (*p* = 0.006 for Nelson) and shyness for Jensen [13] (*p* = 0.05). Other secondary traits are added to it, such as impulsivity (*p* = 0.004) for Lane [16], inflexibility (*p* = 0.033) for Isik [14], approach and withdrawal (*p* = 0.0015) for Lochary [10].

Jensen [13] also found that the mean temperament score of girls was significantly higher than for the boys (*p* = 0.041, independent *t*-test).

## 4. Discussion

The literature on the impact of a child’s temperament on his/her behaviour during a dental procedure is quite sparse and heterogeneous. While some studies have reported an association between various temperaments and DBMP, thus far, no comprehensive qualitative summary of the impact of a child’s temperament on his/her behaviour during a dental procedure has been published.

This literature review was able to highlight some trends. Several temperamental traits seem to be associated with disruptive behaviours that prevent the execution of dental intervention in correct and safe conditions. According to studies using the CBQ/BSQ scale, the dimensions associated with DBMP are the level of **activity, approach and withdrawal, adaptability, the intensity of reactions and attention span**. For studies with the EAS scale, **emotionality** and **shyness** stand out [10,11,12,13,14,15,16,17,18,19]. Among the included studies, only one reported a significantly higher temperament score for girls than for boys [13]. The author explained these findings based on the difference in behaviour characteristics between boys and girls.

**The two main scales used are BSQ and EAS**.

The EAS theory has the advantage of covering relatively consensual dimensions that are found in most current conceptualisations of temperament [8]. The resulting rating scale is the EAS/EASI. It has been adapted for children from the age of one [8] and there is also a version for adolescents and adults.

This tool was the subject of a validation study in France in children aged 6 to 12 in 2002 [4] then aged 2 to 9 in 2013 [24], and the authors were able to verify that the factor structure of the EAS remains the same in these age groups. This finding is important because it offers the possibility of using the same tool to measure temperament in age categories which often require the use of different psychometric measuring instruments depending on the stage of development reached by the child [24].

**These findings resonate with previous studies about temperament and stress resilience**. Resilience as it is applied to children in a stressful environment is often described as the capacity to fight, recover and bounce back in the face of a stressful experience [25]. Several studies have found substantial evidence of a role for childhood temperament in the aetiology and regulation of the child’s emotions. According to these studies, the temperament dimensions associated with behavioural symptoms are: high activity level, low adaptability, withdrawal from new stimuli, distractibility and high intensity [5,26,27,28].

A stressful experience in the dental office can refer to the anticipation, the preconceived fear of pain or possible discomfort during the dental procedure. The same dimensions were observed in the studies included in this review.

**It is interesting to note that these temperament scales have been used in various medical specialties other than child psychiatry (e.g., dental surgery and anaesthesiology in their respective clinical environments) and that they reveal similar results**.

Several studies assessed children’s behaviours in the general medical context with non-invasive procedures requiring only that the child stay calm and stand still, such as radiological examinations (Magnetic Resonance Imagining MRI/Computerised Tomography CT scan) or the preparation before an operation under general anaesthesia. The difficulty of performing these procedures lies in the separation from the parents, as well as the stillness of the child during the radiological examination. The children studied were between 2 and 10 years old. 

While using the EAS scale, Quinonez [29] reports that shyness was linked with disruptive behaviours in awake children during the “pre-separation” phase (*p* = 0.0038) and the “separation” phase (*p* = 0.0281) with the parents [29]. Kain [30] only found a correlation between DBMP and Emotionality (*p* = 0.001). It should also be noted that shyness was not a temperament trait studied by Kain [30].

According to Voepel-Lewis [7], children who experienced failure of conscious sedation by administration of midazolam for radiological examinations were less persevering than those with successful sedation (*p* = 0.05) as well as those who cooperated well while awake for X-ray examinations (*p* = 0.02). They were also more active than the latter (*p* = 0.03).

**The same characteristics of temperament are found in restless children in a dental office, as well as in an operating room or a radiographic exam**.

**These statements also suggest the possibility of extrapolating these scales, as well as their interpretation by specialists other than child psychiatrists in their day-to-day clinical setting**.

More prospective longitudinal studies, with a target age population of children and a larger number of subjects, are needed to confirm these results. In fact, in most of the included studies, the included age groups are wide and cover several periods of the child’s cognitive and emotional development. **It seems important to consider each period of development separately in order to be able to distinguish the dominant temperament dimensions.**

**The main limitation of this review is the diversity of the temperament questionnaires and scales used in the literature**.

Nevertheless, if we compare the two main temperament scales (EAS and BSQ/CBQ) in terms of their questions and items, the same temperament dimensions are found:The approach, the withdrawal and the inability to adapt the BSQ/CBQ correspond to the sociability of EAS.The intensity of the reactions corresponds to the emotionality.The level of motor activity, as well as distractibility, corresponds to hyperactivity.

In spite of the different scales and techniques that are used to determine behaviour disorders and temperament traits in children, the parental rating scales have been successfully used in the studies included and have been valuable for predicting a child’s acceptance of dental treatment.

In addition, the problem of threshold of temperament arises. From what temperament score can we consider the child as hyperactive or shy, and therefore at risk of developing DBMP? The authors do not always provide precise cut-offs, and these may vary from one study to another. **It seems necessary to define specific thresholds according to the chosen temperament scale.**

**Other factors can influence the behaviour of the child, such as anxiety or phobia of the child or his/her parents in relation to dental surgery** [1,2]. Parents, during the interview, may project their own anxiety and convey a negative image of dental care. They may also purposely exaggerate the child’s behaviour in order to be offered an intervention under general anaesthesia, often seen as the easy way out. Faced with this risk of bias, it may be interesting to focus the items of the questionnaire on typical daily situations without any direct link with the dental environment and to mix the items in non-logical order in order to obtain the most objective answer possible.

**A child’s dental anxiety can also influence his/her behaviour**. According to Krikken, aggressive or emotionally reactive children aged between 4 and 12 describe higher dental anxiety [10]. Kalra also studied the impact of dental anxiety and found that children with “difficult”, “active” or “fiery” temperaments, according to the Thomas and Chess classification, have significantly higher pre-extraction anxiety levels [10].

**Finally, the nature of the analysed intervention can have repercussions on the child’s behaviour**. All reviewed studies distinguished between procedures that are defined as invasive and require local anaesthesia and the other non-invasive procedures. Pai dissociated three levels of interventions according to their invasiveness and noticed a significant difference in the behaviour of the child between these acts [17]. **However, to our knowledge, no article assessed the duration of these acts. Would a short act such as an avulsion of temporary teeth be better perceived by the child than an endodontic or restoration procedure? Would an avulsion be more “traumatic”, and therefore less well-perceived than a restoration?**

**The influence of a child’s temperament on oral health is significant. Some authors have also studied this behavioural trait as an indicator of an early childhood decay risk** [17].

This research theme seems important to us, allowing us to refine the selection and the methods of care in efficient and safe conditions. These temperament scales can be useful tools for general practitioners and dental surgeons to identify difficult children who require more in-depth sedation. Future research is needed to ascertain the strength of the findings of the present study and also to investigate other variables related to behavioural difficulties during dental treatment, such as the parental influence on preconceived dental fear.

## Figures and Tables

**Figure 1 children-10-00090-f001:**
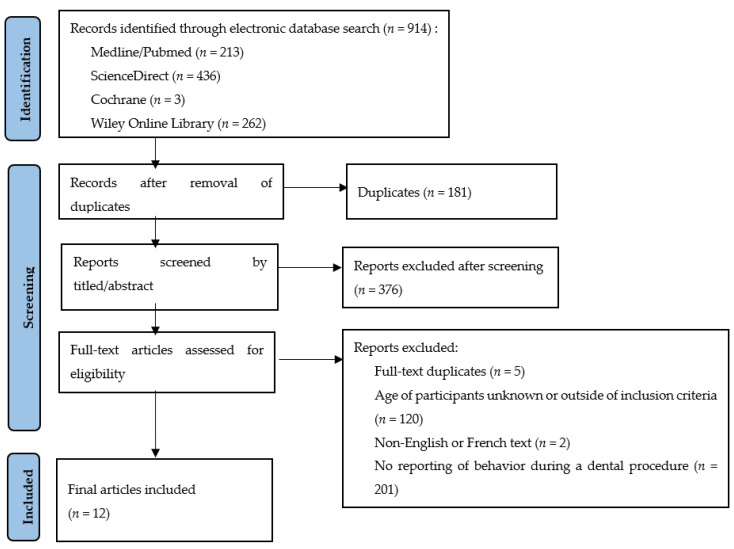
Flow chart of selected articles.

**Table 1 children-10-00090-t001:** Search strategy.

#	Search Term
1	temperament [All Fields] AND dent [All Fields]
2	temperament [MeSH Terms] AND child [All Fields] AND dent [All Fields]
3	child’s temperament [All Fields] AND dent [All Fields]
4	toddler [All Fields] AND temperament [All Fields] AND dent [All Fields]
5	toddler [All Fields] AND temperament [All Fields] AND dent [All Fields]
6	dental behavior [All Fields] AND temperament [All Fields]
7	dental behavior [All Fields] AND temperament [All Fields] AND child [All Fields]
8	behavior [All Fields] AND child [All Fields] AND temperament [All Fields]

**Table 2 children-10-00090-t002:** Summary of the articles included in the literature review.

Author and Year of Publication	Age (Years)	N	Temperament Scale Used	Type of Procedure	Sedation Used	Behaviour Rating Scale	Results: Temperament Statistically Associated with DBMP
Lochary 1993 [10]	18–36 months	29	TTS	Invasive: restoration under local anaesthesia	2 mg/kg oral Hydroxyzine + 2 mg/kg submucosal Meperidine	Ohio State university Behavior Rating Scale (4 items)	**Approach/withdrawal** (*p* = 0.0015) **Adaptability** (*p* = 0.009)
Radis 1994 [11]	3–5	50	BSQ	Non-invasive: initial dental examination + X-rays + prophylaxis	None	Ohio State University Behavior Rating Scale	**Low Approach/withdrawal** (*p* = 0.0023) **Low adaptability**: (*p* = 0.0022) **Intensity/crying**: (*p* = 0.0351) **Activity**: (*p* = 0.0157)
Arnrup 2002 [12]	4–13	203	EASI + Rutter scale	Assessment of factors that influence children cooperation. No information found about the disruptive behaviour nor the dental procedure leading to this disruptive behaviour.			**Emotionality** (*p* < 0.001) **Impulsivity** (*p* < 0.001)
Jensen 2002 [13]	17–51 months (4,5 years)	50	EAS + Shyness	Invasive: avulsion under local anaesthesia	0.3 mg/kg intra-rectal Midazolam	Level of sedation (Wilton) + Child’s acceptance of procedure according to Host	**Shyness** (*p* = 0.05) **Emotionality** (*p* = 0.01) Mood changing
Isik 2010 [14]	4–8	60	STSC (30 items) + CPRS-R (80 items)	Invasive: treatment under local anaesthesia	0.75 mg/kg oral Midazolam + N2O/O2 40/60%	Houpt Sedation Rating Scale (HSRS)	**Inflexibility** (*p* = 0.033)
Aminabadi 2011 [15]	1–7	196	ECBQ for 18–36 months and CBQ for 3–7 years old	Invasive: restoration under local anaesthesia	None	Frankl	**Anger, irritability, fear, reaction, reactivity, shyness**. (correlation coefficient = 0.33 *p* < 0.05)
Lane 2015 [16]	36–95 months (7,9 years)	61	CBQ SF	Invasive: restoration under local anaesthesia	0.3 mg/kg Midazolam + 1 mg/kg hydroxyzine + 1.5 mg/kg Meperidine (50 min latency) + MEOPA	Houpt Sedation Rating Scale (HSRS)	**Impulsivity** (*p* = 0.04)
Pai 2015 [17]	7–11	165	Standardized multi-factor questionnaire including personality	-Non-invasive -Moderately-invasive -Highly invasive (with local anaesthesia)	None	Venham	**Sociability** Interaction with siblings (*p* = 0.00001) Interaction with other children (*p* = 0.00001) Conduct towards parents (*p* = 0.00001) School performance (*p* = 0.0004)
Nelson 2017 [18]	3–5,5 5,5–8	48	CBQ-SF: 3 groups with 15 subgroups, 94 items	Invasive: restoration + tooth extraction under local anaesthesia	N2O/O2 50%	Frankl	**Effortful control** (*p* = 0.001): **Attention control** (*p* = 0.002), **Inhibitory control** (*p* = 0.001) **Negativity Affectivity** (*p* = 0.006): frustration (*p* = 0.006) sadness (*p* = 0.011) soothability (*p* = 0.006) **Extraversion/surgency**: activity (*p* = 0.004), impulsivity (*p* = 0.018)
Tsoi 2018 [19]	4–12	113	EAS	No information	No sedation reported	Frankl	**Emotionality** (r = 0.497 *p* < 0.001) **Activity** (r = 0.196 *p* < 0.03) **Shyness** (r = 0.281 *p* < 0.003)
Jain 2019 [20]	3–5	100	EAS	Non-invasive: initial dental examination + X-rays + prophylaxis	None	Frankl (score of 1–4)	**Emotionality** a(Spearman’s correlation coefficient rs = 0.28) (*p* = 0.046) **Shyness** associated with dental anxiety (rs = 0.28) (*p* = 0.897) **Activity** (*p* = 0.012)
Janeshin 2021 [21]	3–7	215	CBQ	Invasive: dental pulp treatment with local anaesthesia	None	Frankl	Mean scores of fear (*p* = 0.004) and **perceptual sensitivity** (*p* = 0.001) were higher in completely negative behaviour than other temperament traits

## Data Availability

Not applicable.

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
