# Peer review of "Influence of Child’s Temperament on Behaviour Management Problems in the Dental Office: A Literature Review"

_children, 2023, doi:10.3390/children10010090_

Round 1

Reviewer 1 Report

Dear Editor,

I feel honoured with your invitation to review the manuscript entitled "Influence of child’s temperament on behavior management problems in the dental office". The manuscript topic is relevant, novel and well-written. I noted several strengths in the manuscript. The search strategies are appreciated. Although the review article carries much strength and covers a relevant area, I suggest some minor modifications. The following are my comments:

1.   The title and abstract are good. I suggest adding the word 'review' in the title which will attract the readers. As per the guidelines of the journal, the abstract should be fitted with Background, Methods, Results and Conclusion.

2.   The introduction is well-written. “According to the authors” may be replaced with According to research. It should be written in a third form. Please merge one line sentence with the adjacent paragraph. In the sentence “they are individual differences in emotional responsiveness and behavior”, it is not clear to whom the authors refer ‘they’. Please make it clear. The authors may try to sharpen the study arguments by highlighting some implications.

3.   Although the method section covers all required aspects, some clarifications are needed to be added. For example, the author should mention the source/s of choosing the inclusion criteria.

4.   The results are presented well. I suggest merging one sentence with the adjacent paragraph. Years should be added with ‘Aminabadi’ and ‘Janeshin’. The statistical values should be reported as per some standard guidelines throughout the manuscript.

5.   The discussion also needs minor modifications. The sentence ‘According to studies using the CBQ/BSQ scale, the dimensions associated with DBMP are the level of activity, approach and withdrawal, adaptability, the intensity of reactions and attention span. For studies with the EAS scale, emotionality and shyness stand out.’ Needs citations. Small sentences may be merged with adjacent paragraphs. The statistical values should be reported as per some standard guidelines throughout the manuscript. Interpretations of the results need improvements. Novel findings should be highlighted.

In essence, the study topic, write-up, structure and presentations are satisfactory. The study findings may contribute to the current field of the area. As it stands, the manuscript requires minor modifications before it is accepted.

I congratulate the authors for their efforts preparing the manuscript.

Good luck with current and future endeavours.

Author Response

First of all, we would like to thank the reviewer for his work. We always strive to do better, welcome all suggestions and hope that this revised version will be adequate.

Reviewer 2 Report

Reviewer Comments for Manuscript

Influence of child’s temperament on behavior management problems in the dental office

General Comments

Needs major revision.

Needs major English language revision.

What is the rationale of the review.

Specific Comments

Bottom of Form

Title

Suggested title:

Influence of child’s temperament on dental behavior management problems: A literature review

Abstract

Line 2: pls replace the word “article” with “review”.

Line 9: why lit search did not include Scopus, ScienceDirect, Wiley and Google Scholar.

Lines 12-13: CBQ and EAS need to be written in full term followed with the abbreviation.

Line 16: is it methods of sedation or levels of sedation.

Introduction

Line 21: Dental surgeon specialised in paediatric dentistry is considered…. Pls revise and modify to Pediatric dentists are considered…..

Line 25: According to research better than saying According to the authors.

Line 36: “in order to present them with the best sedation” better say “a suitable sedation method”.

Line 46-48: Pls revise language and add more than 1 reference.

Lines 50-51: The objective is somewhat different from the one in the abstract. Pls revise and modify. Also, our objective “was” not “is”.

Materials and Methods

Lines 54-55: Pls add a reference for PRISMA guidelines.

Line 74: abstracts NOT abstract.

Why the databases searched did not include Scopus, ScienceDirect, Wiley and Google Scholar.

Was there any year or language restrictions? Please mention.

Results

Figure 1 need revision. Pls add the number of records excluded after removal of duplicates. Also, in figure 1, abstract and text (Line 92), revise the number of “Final articles included”.

Table 2: Please write the different types of studies from old to recent.

P-value NOT p-value.

Lines: 182-186, 189-194, 287-291, 295-297: Please add reference/s number/s after the author name/s.

Discussion

This section may usefully start with a brief summary of the major findings.

Lines 232-240: All authors names should be followed with reference number. Please revise journal guidelines in reference writing.

Lines 215-220: Second paragraph is already mentioned in the methods section. Pls delete.

P-values are usually mentioned in the results section.

Lines 254-271: The two main scales used are BSQ and EAS. This section needs to be moved to the beginning of the discussion section. Also, please mention limitations towards the end of the discussion.

Implications for practice and implications for research need to be mentioned.

Future directions need to be added.

Please add conclusions and recommendations sections.

References

Also, please check journal guidelines for reference writing.

References needs to be 10 years back not more (from 2012 to 2022).

Please decrease the number of references.

Old references need to be replaced by recent ones.

Ref # 6, the weblink is not working.

Some references have missing information (Ref # 4, 7, 18, 20).

Some of the journals were abbreviated, while others were written in full term (8, 9).

Some of the references include DOI, others do not include DOI number.

In general, all references need to be revised, standardized and written according to the journal guidelines.

Author Response

Article

Influence of child’s temperament on behaviour management problems in the dental office: a literature review

Nhat Minh Do 1, François Clauss 1, Margot Schmitt and Marie-Cécile Manière 1

AUTHOR’S RESPONSE TO REVIEWER

First of all, we would like to thank the reviewer for his work. We always strive to do better, welcome all suggestions and hope that this revised version will be adequate.

Title : modified

Abstract : modified according to comments. The literature search has been updated with the suggested databases.

Introduction : modified according to comments

Materials and Methods : modified according to comments. There was no time restriction for the search. The language restrictions were already mentioned in the eligibility criteria “published in English or French in a peer-reviewed journal”

Results : modified according to comments

Discussion : modified according to comments. The brief summary of the major findings has been added at the start of this section. Implications for practice and future directions has also been added.

References has been revised according to journal guidelines. Some of the references don’t include DOI number because unavailable. PMID number has been added instead. We decided to keep the “old” references because the literature is quite sparse on this subject.

The manuscript was sent for English revisions with a native English-speaking colleague.

The rationale of this review is to gather evidence about the association between the child temperament and his dental behaviour and whether parental temperament ratings of the child can be predictive of his behavioural responses during dental treatment.   

Round 2

Reviewer 2 Report

None

Author Response

Thank you for your comments.

After reviewing our article, we feel that this second version has answered your main concerns and has best represented our work.

If you think it needs a rewrite, please be more specific about the modifications.